# Gestational weight gain in sub-Saharan Africa: Estimation based on pseudo-cohort design

**Samson Gebremedhin**[1]*, **Tilahun Bekele**[2]

**1** School of Public Health, Addis Ababa University, Addis Ababa, Ethiopia, **2** Center for Food Science and Nutrition, Addis Ababa University, Addis Ababa, Ethiopia

* samsongmgs@yahoo.com

## Abstract

### Background

Inadequate or excess gestational weight gain (GWG) leads to multiple undesirable birth outcomes. Yet, in sub-Saharan Africa (SSA) little is known about the weight gain pattern in pregnancy. The purpose of the study is to estimate the average gestational weight gain (GWG) in sub-Saharan Africa (SSA) and to examined whether there had been recent improvements or not.

### Methods

Based on cross-sectional anthropometric data extracted from multiple Demographic and Health Surveys conducted in SSA, we estimated the average GWG in the region. Pseudo-cohort design was used to reconstruct GWG trajectories based on aggregated data of 110,482 women extracted from 30 recent surveys. Trend in GWG between 2000 and 2015 was determined using the data of 11 SSA countries. Pre-pregnancy weight was estimated based on the weight of non-pregnant women at risk of conception.

### Results

On average, women in SSA gain inadequate weight (6.6 kg, 95% confidence interval, 6.0–7.2) over pregnancy. No meaningful gain was observed in the first trimester; whereas, women in the second and third trimesters put on 2.2 and 3.2 kg, respectively. The highest weight gain (10.5, 8.2–12.9 kg) was observed in Southern African sub-region and the lowest in Western Africa (5.8, 5.0–6.6 kg). The GWG among women who had secondary or above education (9.5, 8.2–10.9 kg) was higher than women with lower education (5.0, 4.3–5.8 kg). Likewise, GWG in women from richest households (9.0, 7.2–10.7 kg) was superior to those from poorest households (6.1, 5.3–7.0 kg). The estimated recent (2015–20) mean GWG (6.6, 5.8–7.4 kg) was not significantly different from what had been at beginning of the new millennium (6.7, 5.9–7.5 kg).

### Conclusion

In SSA GWG is extremely low and is not showing improvements.

**Data Availability Statement:** The analysed data is publicly available from https://dhsprogram.com/data/.

**Funding:** The authors received no specific funding for this work.

**Competing interests:** The authors have declared that no competing interests exist.

## Background

Gestational weight gain (GWG)–the weight increase between conception and just before the birth of the baby–is an important predictor of maternofoetal nutrition [1]. Data on GWG is imperative because insufficient gain is a proxy indicator for maternal and foetal undernutrition and shows strong correlation with intrauterine growth retardation and low birthweight [1, 2]. Low GWG may also increase risk of maternal and perinatal death [3]. On the other hand, excessive GWG leads to increased risks of macrosomia, caesarean delivery, preeclampsia, childhood obesity and maternal postpartum weight retention [4–7].

In 2009 the Institute of Medicine (IOM) of the National Academies put forth a new guideline on rate of GWG [8]. Though the guideline had been developed for the US population, the available few studies suggested its potential applicability in other settings as well [9–11]. According to the guideline, women with normal pre-pregnancy body mass index (BMI) and having singleton pregnancy should put on 11.5–16.0 kg of weight, assuming 0.5–2 kg gain in the first and 0.42 kg/week in the last two trimesters. The gain should also be as high as 18 kg in women with low pre-pregnancy BMI [8]. GWG is affected by multiple factors including socioeconomic status, dietary intake, co-morbidities, multifetal pregnancy and genetic factors [12].

In many low-income settings including the sub-Saharan African (SSA) limited information is available on the rate of GWG and there is no GWG guideline specific to the region [9, 13]. This is probably because estimation of GWG requires women to be followed from pre-pregnancy to near to childbirth, which is frequently infeasible in low-income settings where preconception care is rudimentary and births are largely happening at home [14, 15]. However, the available limited studies reported high rates of inadequate GWG in Africa and Asia [9, 13]. A systematic review concluded that 31% of women in Asia, as compared to 18–21% in Europe and US, gain inadequate gestational weight [10]. A study that reviewed the available few studies in SSA reported that the prevalence of optimal GWG ranged from 3% to 62% [13].

A recent study by Wang and colleagues (2020) based on modelling of Demographic and Health Surveys (DHS) reported that in 2015, the mean GWGs in Latin America and Caribbean region (11.8 kg) and Central Europe and Eastern Europe (11.2 kg) was considerable higher than the corresponding estimates for Sub-Saharan Africa (6.6 kg) and North Africa and Middle East (6.8 kg) [16]. However, this study did not provide information on the trends in GWG in the region, sub-national estimates and differences in GWG trajectories across basis maternal socio-demographic characteristics.

The purpose of the current study is to: (i) estimate mean GWG in SSA based on aggregated data from multiple nationally representative cross-sectional surveys and; (ii) to compare changes in mean GWG between 2000 and 2015 in the region. Information on the extent of GWG important because weight gain rate is a proxy indicator for maternal nutrition and helps to monitor the indicator for informing maternal nutrition related services and programs.

## Methods

### Study design

Pseudo-cohort design was used to estimate GWG based on aggregated cross-sectional anthropometric data of pregnant and non-pregnant women enrolled in recent and nationally representative DHS. Weight gain trajectories during pregnancy were reconstructed using the data of pregnant women at different gestational durations (1–9 months) whereas pre-pregnancy weight was estimated using the data of non-pregnant women at risk of conception.

### Data source and inclusion criteria

The analysis was made based on the secondary data of Standard DHS. Other types of DHS including Continuous DHS, Malaria Indicator Surveys and AIDS Indicator Surveys that normally do not collected anthropometric data were not considered.

In order to estimate mean GWG, we analysed the data of 110,482 women in reproductive age (19,850 pregnant and 90,642 non-pregnant women) enrolled in DHS implemented in 30 SSA countries since 2010. With the intension of limiting the study to the recent period, surveys conducted before 2010 were excluded. At times when two or more eligible surveys were available for a country, the most recent was considered.

In order evaluate trends in GWG between 2000 and 2015, we compared the data of 11 SSA countries that implemented DHS around the beginning of the new millennium (1997–2003) and on or after 2015 (2015–2018). We were not limited to studies conducted in 2000 and 2015, rather we analysed studies conducted around these two periods because very few countries implemented surveys exactly on 2000 and 2015. In this specific analysis the data of 54,603 subjects (9,508 pregnant and 41,095 non-pregnant women) from surveys implemented between 1997 and 2003 and the data 49,127 subjects (9,055 pregnant and 40,072 non-pregnant women) from surveys taken place after 2014 were compared (Table 1).

S1 File outlines the process of refining the dataset for the final analysis. Regarding pregnant women, data lines having complete information on gestational age and body weight were eligible for analysis. For non-pregnant women those at risk of conception were eligible. Women at risk of conception was operationally defined as menstruating women in reproductive age who were not abstaining from sex and not using any modern or natural contraceptive methods at the time of the survey. Furthermore, women who recently gave birth (in the last one year) were excluded from the analysis (S1 File).

### Sampling design and approach of data collection in DHS

Demographic and Health Surveys are standardized national surveys being regularly implemented by the DHS Program and national agencies in many low- and middle-income countries for monitoring vital statistics and population health indicators [17]. The DHS identify eligible subjects including women 15–49 years of age, using two-stage multiple cluster sampling approach that is intended to provide representative data at national and subnational (regions or states) levels in both urban and rural settings [18].

In DHS socio-demographic data are collected using standardized questionnaires and anthropometric (weight and height) measurements are taken following standard procedures. Pregnancy status is determined based on self-report without any further validation. Gestational duration is determined in months (1–9 months) based on self-report and the date of last normal menstrual period (LNMP) and is approximated to the nearest month.

### Estimation of pre-pregnancy weight

Pre-pregnancy weight was estimated based on the mean body weight of women at risk of conception. Sexually active and menstruating women, who were not using any contraceptive at the time of the survey were considered to be at risk of conception. In this regard, the data of women who gave birth in the last one year was not used considering that postpartum weight retention can make us to overestimate pre-pregnancy weight. The weight of women at risk of conception, rather than that of non-pregnant women was used because the latter are likely to be systematically different from pregnant women in basic socio-demographic characteristics.

**Table 1. Demographic and health surveys included in the analysis.**

| Country | Year of survey | | | |
| --- | --- | --- | --- | --- |
| | Surveys used for estimating mean GWG | | Control surveys for assessing trends in GWG | |
| | Year | Sample size‡ | Year | Sample size‡ |
| Benin | 2017/18 | 4,394 | 2001 | 3,655 |
| Burkina Faso | 2010 | 4,306 | - | - |
| Burundi | 2016/17 | 4,695 | - | - |
| Cameroon | 2018 | 3,879 | 1998 | 512 |
| Chad | 2014/15 | 6,451 | 1996/97 | 1,926 |
| Comoros | 2012 | 3,444 | - | - |
| Congo | 2011/12 | 1,898 | - | - |
| Congo Democratic Republic | 2013/14 | 4,636 | - | - |
| Cote d'Ivoire | 2011–12 | 2,625 | - | - |
| Ethiopia | 2016 | 7,275 | 2000 | 10,302 |
| Gabon | 2012 | 2,725 | - | - |
| Gambia | 2013 | 2,738 | - | - |
| Ghana | 2014 | 2,622 | - | - |
| Guinea | 2018 | 2,891 | 1999 | 1,358 |
| Kenya | 2014 | 6,134 | - | - |
| Lesotho | 2014 | 2,292 | - | - |
| Liberia | 2013 | 1,235 | - | - |
| Malawi | 2015/16 | 2,705 | 2000 | 7,048 |
| Mali | 2018 | 3,107 | 2001 | 7,504 |
| Namibia | 2013 | 1,504 | - | - |
| Niger | 2012 | 2,752 | - | - |
| Nigeria | 2018 | 8,177 | 2003 | 4,792 |
| Rwanda | 2014/15 | 3,459 | 2000 | 6,788 |
| Senegal | 2010/11 | 3,500 | - | - |
| Sierra Leone | 2013 | 3,845 | - | - |
| South Africa | 2016 | 1,324 | - | - |
| Tanzania | 2015/16 | 6,458 | - | - |
| Togo | 2013/14 | 2,622 | - | - |
| Uganda | 2016 | 2,829 | 2000/01 | 3,624 |
| Zimbabwe | 2015 | 3,960 | 1999 | 3,094 |
| Total | | 110,482 | | 50,603 |

‡ sample size for both pregnant and non-pregnant women; GWG–Gestational weight gain.

### Reconstructing weight gain patterns during pregnancy

Gestational weight gain during pregnancy was reconstructed by taking the mean weight of pregnant women at different gestational months (1–9 months) and assuming that pregnant women in the region have passed through the month-specific weight gain trajectories.

### Estimation of weight at the time of delivery

As described earlier, in DHS gestational duration is measured to the nearest month. Accordingly, maternal weight at a given month represents the average gestational weight in the +/- 2 gestational weeks range. Accordingly, the average gestational weight at the 9 month underestimates maternal weight at the time of birth. According maternal weight at the time of delivery

was estimated by extrapolating the weight gain rate in the third trimester (6–9 months) to the estimated date of birth (40th week or 9 months and 10 days).

## Data management and analysis

We used SPSS v24 for data management and analysis. The datasets for all the eligible surveys were downloaded from the DHS Program database [19] and merged into a spreadsheet.

Weighted data analysis was employed based on the sampling weights readily available in the data and post-stratification weight developed using the 2020 population size of the countries [20]. DHS calculates sampling weights based on sample selection probabilities at household and individual levels [18]. Post-stratification weight was computed by dividing the 2020 total population size of that country to the total population size of the countries represented in the dataset for the same year. The ultimate data weight was calculated by multiplying the sample weight readily available in the dataset by the post-stratification weight with linearization to balance the weighted and unweighted sample size. The analysed data is publicly available from https://dhsprogram.com/data/.

When available, gestational age was determined based on LNMP otherwise it was estimated based on self-report of the women. Very small proportion (about 0.3%) of pregnant women reported gestational age of 10 months and during analysis it was recoded to 9 months.

Average gestational weight gain was computed by subtracting estimated pre-pregnancy weight with estimated weight at the time of delivery. Weight gain in the first trimester was estimated by subtracting pre-pregnancy weight from estimated weight at the third gestational month. Second trimester weight gain was computed by subtracting weight at third gestational month from weight at the sixth month. Similarly, third trimester weight gain was computed by subtracting weight at sixth gestational month from estimated weight at the time of delivery.

Total GWG and trimester-specific weight gain rates were estimated at regional and sub-regional levels However, country-level estimates have not been provided as many national surveys enrolled inadequate number of pregnant women at different gestational durations. Sub-regional estimates are provided by classifying the region into four geographic areas (Eastern, Southern, Western and Central) according to the African Union classification system [21]. Furthermore, the countries were sub-divided as low-income ($1,035 or less) or lower-middle or upper-income ($1,036 to $12,535) economies based the classification approach proposed by the World Bank [22].

GWG rates were compared across levels of maternal age, place of residence (urban vs rural), maternal educational status, household wealth index and maternal height. Wealth index was computed as an index of household economic standing using Principal Component Analysis based on ownership of selected household assets and materials used for house construction [18]. For comparing weight gain trajectories across levels of the aforementioned variables, 95% confidence intervals (CIs) for mean GWG were used.

## Results

### Basic characteristics

The data of 88,602 non-pregnant and 21,822 pregnant women from 30 countries including 13 Western, 6 Eastern, 6 Central and 5 Southern African countries, was used for estimating mean GWG in SSA. The mean (± SD) age of the women was 26.9 (± 10.1) years and nearly one-third (30.2%) were under the age of 20 years. Two thirds were rural residents and one third had no formal education. The number of pregnant women across the gestational months ranged from 934 in the first to 3,204 in the eighth month. Women in the first trimester were slightly under-represented (25.6%) while women in the second trimester were overrepresented (40.5%).

About a quarter (26.8%) the women were stunted (height < 155 cm) and among non-pregnant women 19 years or above 12.1% were underweight and 24.4% were overweight/obese (Table 2).

## Estimated gestational weight gain in sub-Saharan Africa

Fig 1 depicts the GWG trajectory in SSA. GWG increased by 6.6 kg (95% CI: 6.0–7.2) from the estimated pre-pregnancy weight of 57.1 kg (95% CI: 57.0–57.2) to 63.7 kg (95% CI: 63.0–64.4) at the end of pregnancy. No meaningful weight gain was observed in the first trimester; whereas, average gains were 2.2 kg in second and 3.2 kg in the third trimesters, respectively (Fig 1).

Table 3 compares the estimated pre-pregnancy weight and GWG in different geographic regions and economic categories of 30 SSA countries. The Southern Africa (59.8 kg: 95% CI: 59.5–60.1) and Eastern Africa (55.2 kg: 95% CI: 55.0–55.3) sub-regions had the highest and lowest pre-pregnancy weight, respectively. On the other hand, the highest GWG (10.5 kg: 95% CI: 8.2–12.9) was in Southern African sub-region and the lowest was in Western Africa (5.8 kg: 95% CI: 5.0–6.6). In Central and Eastern regions, the estimated GWG rates were 6.8 kg (95% CI: 5.4–8.1) and 6.6 kg (95% CI: 5.5–7.6), respectively.

The estimated GWG in low-income countries was 6.4 kg (95% CI: 6.1–7.7) while the corresponding level for lower-middle or upper-income countries was 6.8 kg (95% CI: 5.8–7.8). The overlap in the confidence intervals suggested absence of statistically significant difference between the two groups (Table 3).

The GWG pattern was also compared across selected socio-demographic characteristics. The weight gain among women who had secondary or above level of education (9.5 kg: 95% CI: 8.2–10.9) was significantly higher than women who had no formal education (5.0 kg: 95% CI: 4.3–5.8). Similarly, the GWG among women from richest households (9.0 kg: 95% CI: 7.2–10.7) was also superior to those from poorest households (6.1 kg: 95% CI: 5.3–7.0). No significant differences were observed across levels of maternal age, place of residence and maternal height (Table 3).

## Trends in gestational weight gain

Fig 2 compares the GWG trajectory in 11 SSA countries between the beginning of the new millennium (1997–2003) and after 2014. Over the period, the pre-pregnancy weight significantly increased from 54.4 kg (95% CI: 54.3–54.5) to 56.9 kg (95% CI: 56.8–57.1). Nevertheless, the GWG remains more or less the same: 6.7 kg (95% CI: 5.9–7.5) and 6.6 kg (95% CI: 5.8–7.4), respectively for the two periods (Fig 2).

## Discussion

In SSA information on GWG remains scarce. This study, based on aggregated cross-sectional data from multiple national surveys, reconstructed weight gain trajectories during pregnancy and estimated that mean GWG is very low in the region and did not show meaningful changes over the last 15 years. Furthermore, GWG showed significant heterogeneity across the sub-regions of SSA and levels of socio-economic status including maternal educational status and household wealth index.

According to IOM, pregnant women with normal pre-pregnancy BMI should gain 11.5–16.0 kg of weight and women with low pre-pregnancy BMI are expected to gain as high as 18 kg of weight [8]. Though the utility the IOM recommendation in settings outside the United States can still be contended [23], the mean GWG that we estimated (6.6 kg) is only about half of the minimum recommended gain for women with normal baseline BMI. A study that

**Table 2. Basic characteristics of the study subjects, sub-Saharan Africa, 2010–2018.**

| Characteristics (n = 110,428) | Frequency | Percentage |
|---|---|---|
| Pregnancy status | | |
| Pregnant | 21,822 | 19.8 |
| Non-pregnant | 88,606 | 80.2 |
| Gestational trimester (n = 21,822) | | |
| First (1–3) | 5,682 | 25.6 |
| Second (4–6) | 8,897 | 40.5 |
| Third (7–9) | 7,399 | 33.7 |
| Gestational month (n = 21,822) | | |
| 1 | 934 | 4.3 |
| 2 | 2,070 | 9.5 |
| 3 | 2,638 | 12.1 |
| 4 | 2,640 | 12.1 |
| 5 | 3,182 | 14.6 |
| 6 | 3,011 | 13.8 |
| 7 | 3,104 | 14.2 |
| 8 | 3,204 | 14.7 |
| 9 | 1,038 | 4.8 |
| Type of place of residence | | |
| Urban | 38,085 | 34.5 |
| Rural | 72,343 | 65.5 |
| Maternal age | | |
| 15–19 | 33,309 | 30.2 |
| 20–34 | 49,402 | 44.7 |
| 35–49 | 27,717 | 25.1 |
| Level of education (n = 110,417) | | |
| No education | 37,487 | 33.9 |
| Primary | 35,962 | 32.6 |
| Secondary | 31,562 | 28.6 |
| Higher | 5,406 | 4.9 |
| Wealth index | | |
| Poorest | 18,201 | 16.5 |
| Poorer | 19,075 | 17.3 |
| Middle | 20,739 | 18.8 |
| Richer | 22,836 | 20.7 |
| Richest | 29,577 | 26.8 |
| Maternal height (n = 110,243) | | |
| < 145 cm | 2,030 | 1.8 |
| 145–154 cm | 27,612 | 25.0 |
| 155 or above | 80,601 | 73.0 |
| Body mass index in non-pregnant women 19 years or above (n = 62,011) | | |
| Underweight | 7484 | 12.1 |
| Normal | 39380 | 63.5 |
| Overweight or obese | 15,148 | 24.4 |
| Mean (± standard deviation) weight (kg) | | |
| Non-pregnant | 57.1 (± 12.7) | |
| First trimester (1–3) | 57.0 (± 11.0) | |
| Second trimester (4–6) | 59.4 (± 11.0) | |
| Third trimester (7–9) | 62.1 (± 11.2) | |

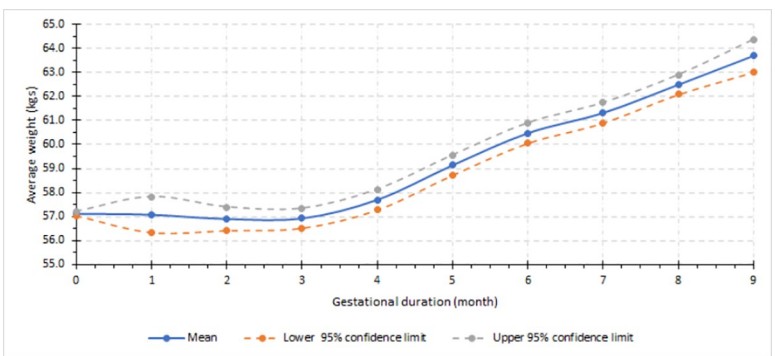

**Fig 1. Gestational weight gain trajectories in sub-Saharan Africa.**

estimated GWG in SSA and India based on surveys conducted between 2000 and 2010 also reported GWG in both regions is grossly inadequate at 7 kg [24]. A systematic review also found that the prevalence of inadequate weight gain exceeded 50% in most of the studies from

**Table 3. Gestational weight gain by household and maternal characteristics, sub-Saharan Africa, 2010–2018.**

| Characteristics | Estimated mean weight (kg) (95% CI) | | | | Estimated gestational weight gain (kg) (95% CI) |
|---|---|---|---|---|---|
| | Pre-pregnancy | Third month | Sixth month | End of pregnancy | |
| Geographic classification of the country | | | | | |
| Southern | 59.8 (59.5–60.1) | 59.6 (58.2–61.1) | 64.4 (62.7–66.1) | 70.3 (67.7–73.0) | 10.5 (8.2–12.9) |
| Central | 56.1 (55.9–56.3) | 56.1 (55.2–56.9) | 59.3 (58.4–60.1) | 62.9 (61.3–64.5) | 6.8 (5.4–8.1) |
| Eastern | 55.2 (55.0–55.3) | 55.8 (55.0–56.6) | 59.8 (59.1–60.6) | 61.7 (60.6–62.9) | 6.6 (5.5–7.6) |
| Western | 57.8 (57.7–58.0) | 57.6 (56.9–58.2) | 60.7 (60.1–61.4) | 63.6 (62.7–64.6) | 5.8 (5.0–6.6) |
| Economic classification of the country | | | | | |
| Low-income | 55.4 (55.3–55.5) | 55.5 (55.0–56.0) | 58.6 (58.1–59.1) | 62.2 (61.3–63.0) | 6.4 (6.1–7.7) |
| Lower-middle-income | 58.9 (58.8–59.1) | 58.6 (57.9–59.3) | 62.7 (62.0–63.4) | 65.7 (64.6–66.9) | 6.8 (5.8–7.8) |
| Maternal age (years) | | | | | |
| 15–19 | 52.5 (52.4–52.6) | 53.7 (53.0–54.4) | 56.8 (56.1–57.6) | 59.4 (58.0–60.7) | 6.9 (5.7–8.1) |
| 20–34 | 57.8 (57.7–57.9) | 57.1 (56.6–57.6) | 60.9 (60.3–61.4) | 64.1 (63.2–64.9) | 6.2 (5.5–6.9) |
| 35 or above | 58.7 (58.5–58.9) | 59.2 (57.9–60.4) | 62.4 (61.2–63.5) | 65.2 (63.4–67.0) | 6.5 (4.8–8.1) |
| Place of residence | | | | | |
| Urban | 60.4 (60.2–60.5) | 60.2 (59.3–61.1) | 65.0 (64.2–65.9) | 67.9 (66.6–69.3) | 7.6 (6.4–8.7) |
| Rural | 55.5 (55.4–55.6) | 55.6 (55.2–56.0) | 58.4 (58.0–58.9) | 61.7 (60.9–62.4) | 6.2 (5.5–6.8) |
| Educational status | | | | | |
| No formal education | 55.8 (55.7–55.9) | 55.1 (54.6–55.7) | 58.2 (57.6–58.7) | 60.8 (59.9–61.7) | 5.0 (4.3–5.8) |
| Primary education | 56.3 (56.2–56.4) | 56.2 (55.5–56.9) | 60.0 (59.3–60.7) | 63.8 (62.6–65.0) | 7.6 (6.5–8.6) |
| Secondary education | 58.8 (58.6–58.9) | 60.6 (59.7–61.5) | 64.6 (63.7–65.6) | 68.3 (66.8–69.8) | 9.5 (8.2–10.9) |
| Household wealth index | | | | | |
| Poor | 53.5 (53.3–53.6) | 54.6 (53.9–55.3) | 56.7 (56.0–57.4) | 59.6 (58.6–60.7) | 6.1 (5.3–7.0) |
| Poorer | 54.9 (54.7–55.1) | 55.7 (55.0–56.4) | 59.4 (58.6–60.1) | 61.8 (60.6–63.0) | 6.9 (5.9–8.0) |
| Middle | 56.4 (56.3–56.6) | 56.2 (55.3–57.0) | 59.0 (58.2–59.8) | 63.2 (61.7–64.7) | 6.8 (5.5–8.1) |
| Richer | 58.2 (58.1–58.4) | 57.7 (56.6–58.7) | 62.9 (61.9–63.9) | 66.3 (64.4–68.1) | 8.0 (6.4–9.7) |
| Richest | 60.7 (60.5–60.9) | 62.3 (61.0–63.6) | 66.7 (65.4–67.9) | 69.7 (67.8–71.6) | 9.0 (7.2–10.7) |
| Maternal height | | | | | |
| < 155 cm | 51.1 (50.9–51.2) | 51.8 (51.1–52.5) | 54.2 (53.6–54.9) | 57.4 (56.2–58.6) | 6.3 (5.3–7.3) |
| > = 155 cm | 58.9 (58.8–59.0) | 58.6 (58.2–59.1) | 62.5 (62.0–63.0) | 65.2 (64.4–66.0) | 6.3 (5.6–7.0) |

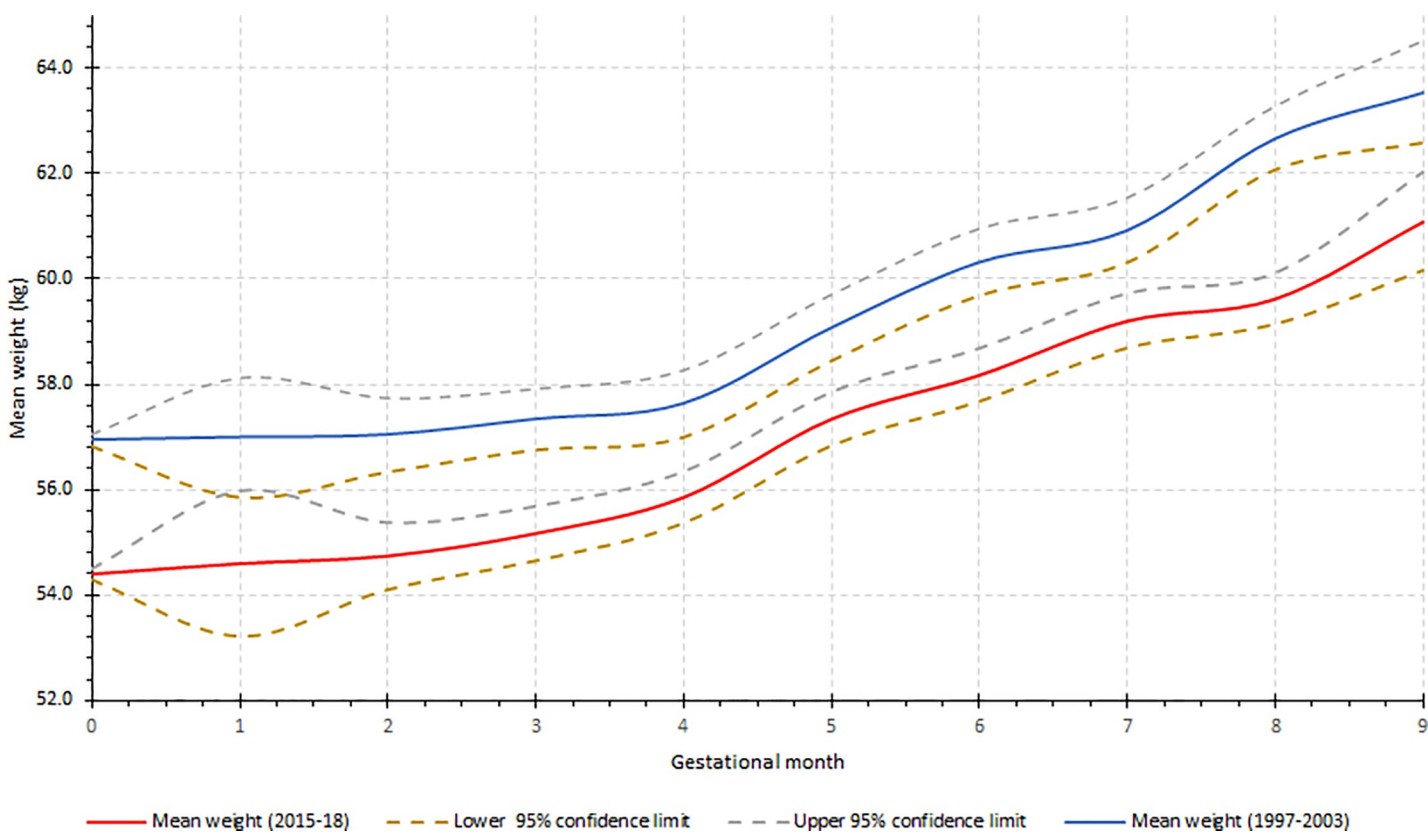

**Fig 2. Comparison of gestational weight gain in sub-Saharan Africa between 1997–2003 and after 2014.**

SSA [13]. The low GWG trajectory observed in the region is reflective of several underlying determinants including suboptimal maternal nutrition, poor access to social services and low socio-economic status [12]. Genetic factors can also significantly affect GWG [12, 25].

Our analysis suggested that GWG in SSA is only increased by 6.6 kg from the estimated pre-pregnancy weight of 57.1 kg to 63.7 kg at the end of pregnancy. This is in conformation with the findings of a recent modelling analysis of DHS by Wang and colleagues [15]. The study reported that in 2015, the estimated GWG in SSA was 6.6 kg, which was much lower than that of Latin America and Caribbean (11.8 kg), Central and Eastern Europe (11.2 kg) and Central Asia (11.2 kg). On the other hand, the GWG in North Africa and Middle East found to be grossly inadequate (6.8 kg). In general, our study and that of Wang et al [15]. followed similar pseudo cohort design to estimate the mean total GWG. Yet, the statistical analysis approaches employed were different.

Sub-regional analysis also suggested that the mean GWG in southern African region (10.5 kg) is considerably higher than the eastern (6.6 kg) and western (5.8kg) sub-regions. This is likely the indirect reflection of the better socioeconomical status of southern African countries like Namibia and South Africa included in the analysis.

We also observed that over the last 15 years of so, pre-pregnancy weight has significantly increased on average by 2.4 kg in SSA yet GWG showed no significant improvement. The finding looks paradoxical because the better nutrition that resulted in improvements in pre-pregnancy weight should also uplift GWG. A study that compared the GWG trajectories among Indians and Africans also reported that both regions have similar GWG despite the pre-pregnancy weight was substantially lower by 8 kg among Indian women [24]. Interventional and

prospective studies are required to evaluate why GWG is not improving in SSA despite nutritional improvements observed over the last two decades.

In the current study better maternal education and household wealth standing were significantly associated with higher rates of GWG. The same had been witnessed by other studies conducted in low- or middle-income countries. In Brazil, pregnant women with seven or less years of schooling were two times more likely to encounter insufficient weight gains as compared to those with better education; however, significant difference have not observed between categories of economic class [26]. In Ilam province of Iran, prevalence of inadequate weight gain was higher among women with lower educational and income status [27]. Better socioeconomic status is likely to increase GWG though advancing access to better nutrition and social services.

The findings of the study should be interpreted in consideration of the following strength and limitations. The major strength is that GWG was estimated based on data coming from several nationally surveys and this assures the representativeness of the findings at regional and sub-regional levels. Conversely, the following limitations have to be noted. (1) We estimated GWG using aggregated cross-sectional, rather than the individual-based longitudinal data. Consequently, prevalence of inadequate or excess weight gain could not be determined and the heterogeneity observed across levels of socio-economic variables could not be statistically adjusted for potential confounders. In addition, GWG estimates cannot be provided for different levels of pre-pregnancy BMI. This was because BMI values were not calculated for pregnant women and reconstructing GWG based on BMI levels was not possible. (2) We determined mean GWG assuming normal gestation length but in actual sense GWG is directly affected by gestational duration [12] and significant proportion (about 15% in total) of pregnancies end up in either pre- or post-term births [28, 29]. This may have resulted in over- or under-estimation of GWG. (3) DHS determine pregnancy status and gestational duration based on self-reports without any further validation. Future more, gestational duration is measured in months rather than weeks. These may have made us to over- or under-estimate the extent of GWG in SSA.

Furthermore, we assessed adequacy of GWG considering the US IOM guideline as a standard. Though the applicability of the IOM standard outside US especially in Asia had been reported [9–11], its usability in African setup had not been investigated. We used the IOM guidelines as a standard because we have not come across with SSA-specific GWG guideline.

While we estimate the trends in GWG between 2000 and 2015 we encountered with the problem that nearly all countries do not have data in these specific years. Accordingly, we have included surveys conducted around 2000 (1997–2003) and 2015 (2015–2018). This decision may have made us to marginally over or underestimate the rate of GWG between 2000 and 2015.

In the current study the presence or absence of statistically significant differences between two mean GWG trajectories was assessed using the overlap between the two associated confidence intervals. We were not able to calculate p-values because estimation of GWG was not made based on individual-level data. While the method of examining differences based on overlap is simple and convenient it may lead to type II error especially when the associated p-values are marginally significant [30, 31].

Though 30 of the 46 SSA are represented in the analysis, the generalizability of the findings to the entire region can still be doubted. This is because there is no guarantee that countries that recently implemented DHS and included in the analysis are comparable with other countries in terms of pertinent variables that many affect GWG. Furthermore, we could not be able to provide country-level estimates because reconstructing GWG trajectories using aggregated data requires large sample size and many of the national surveys enrolled inadequate number of pregnant women.

## Conclusion

The mean GWG in SSA is very low and did not show meaningful changes over the last 15 years. Women from southern African region tend to have better weight gain trajectories than the other sub-regions of the continent. Similarly, women with lower socio-economic status tend to put on smaller weights during pregnancy.

## Supporting information

**S1 File. Flow chart of the study.**
(DOCX)

## Acknowledgments

The authors acknowledge the DHS Program for making the DHS data accessible for public use.

## Author Contributions

**Conceptualization:** Samson Gebremedhin, Tilahun Bekele.

**Formal analysis:** Samson Gebremedhin, Tilahun Bekele.

**Methodology:** Samson Gebremedhin.

**Writing – original draft:** Samson Gebremedhin.

**Writing – review & editing:** Samson Gebremedhin, Tilahun Bekele.

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
