## [Decision Letter · Decision Letter 0]

1 Mar 2021

PONE-D-21-02601

Gestational weight gain in sub-Saharan Africa: Estimation based on pseudo-cohort design

PLOS ONE

Dear Dr. Gebremedhin,

Thank you for submitting your manuscript to PLOS ONE. After careful consideration, we feel that it has merit but does not fully meet PLOS ONE’s publication criteria as it currently stands. Therefore, we invite you to submit a revised version of the manuscript that addresses the points raised during the review process.

The paper looks promising but it currently is lacking in several respects.

With respect to the introduction, the paper should provide background that puts the manuscript into context and allows readers outside the field to understand the purpose and significance of the study and include a brief review of the key literature. In the present case, as reviewer 1 points out, the article is missing a key reference that looks at the same topic with the same data and similar methods but for more countries. It is basic in this case to note the differences in approach or in scope. Note that otherwise PLOS ONE policy states that if a submitted study replicates or is very similar to previous work, authors must provide a sound scientific rationale for the submitted work and clearly reference and discuss the existing literature. Submissions that replicate or are derivative of existing work will likely be rejected if authors do not provide adequate justification.

Regarding methods, there are some limitations that should be addressed.

A prior consideration, is that the methods should be adequately described to allow for replication. As noted by reviewer 1, this is not the case, and the missing information might have confused, for instance, reviewer 3 in understanding the method.

First, note the concerns regarding the use of relevant guidelines as suggested by reviewers 1, 2 and 3.

Second, the problem of the relevant comparison group raised by reviewers 2 and 3. Reviewer 2 suggestion of reweighting non-pregnant women to match pregnant women according to weight is interesting and doable. If you carry this out show, at least for some instances, both sets of estimates to see if the strange patterns for first trimester disappear. Alternatively, a more sophisticated approach would be the use of propensity score matching to ensure that women in the synthetic control group (nonpregnant) are comparable to those pregnant.

Comment by reviewer 3 regarding grouping by weight I don’t believe it is feasible since you do not observe prepregnancy weight. However, you should bear it in mind when analyzing the patterns of estimated gestational weight gain. Since you have computed the proportions underweight and overweight you could see if the pattern of estimated weight gain at the survey level correlate with the proportions underweight and overweight. That will be very helpful for the discussion and also to tease out possible factors behind the much smaller GWG in Africa compared to other developing regions according to the missing source noted by reviewer 1.

Third, as noted by reviewer 2, not using data from women in the first two months of pregnancy is probably a good idea. For instance, it is known that DHS patterns of reported pregnancy termination are extremely low in most sub-Saharan African countries (https://doi.org/10.1371/journal.pone.0221178) suggesting that many early terminations are missing because the woman was unaware of the pregnancy. Since this proportion would vary from survey to survey and according to socioeconomic characteristics, it would induce a bias in estimate weight gain.

In your revision note that PLOS ONE endorses the use of the STROBE checklist (http://www.strobe-statement.org) for observational studies such as this one. Make sure that all the questions in the checklist are addressed.

We look forward to receiving your revised manuscript.

Kind regards,

José Antonio Ortega, Ph.D.

Academic Editor

PLOS ONE

Journal Requirements:

Reviewers' comments:

Reviewer's Responses to Questions

**Comments to the Author**

1. Is the manuscript technically sound, and do the data support the conclusions?

Reviewer #1: Partly

Reviewer #2: Partly

Reviewer #3: Partly

2. Has the statistical analysis been performed appropriately and rigorously? 

Reviewer #1: I Don't Know

Reviewer #2: Yes

Reviewer #3: I Don't Know

3. Have the authors made all data underlying the findings in their manuscript fully available?

Reviewer #1: Yes

Reviewer #2: Yes

Reviewer #3: No

4. Is the manuscript presented in an intelligible fashion and written in standard English?

Reviewer #1: Yes

Reviewer #2: Yes

Reviewer #3: Yes

5. Review Comments to the Author

Reviewer #1: In this study by Gebremedhin and Bekele, the authors used data from the Demographic and Health Surveys (DHS) program to estimate the mean gestational weight gain (GWG) in sub-Saharan African countries. The authors reported that the level of GWG in sub-Saharan Africa was low and did not improve with time.

The use of nationally representative data and the extensive subgroup analyses are the primary strengths of this work. The limitations of this study were the use of cross-sectional data and the inability to estimate the prevalence of women with inadequate or excessive GWG. The manuscript is well-written in general, but I find that the statistical methodology of the analysis is inadequately described. Please find my specific questions, comments, and suggestions below.

1) This work is similar to a previous study in terms of the research question. In Wang et al. (Gestational weight gain in low-income and middle-income countries: a modelling analysis using nationally representative data. BMJ global health. 2020), a multilevel modeling framework was applied to the DHS data to estimate the mean GWG in the year 2015 in all low- and middle-income countries and regions of the world, including in sub-Saharan Africa. Surprisingly, the Wang et al. paper was not referenced or mentioned in this work. Discussing the similarities and differences in the results and methodology between the two works will be necessary.

2) On page 4: “assuming 0.5-2 kg gain in the first and 0.45 kg/week in the last two trimesters." However, the weekly weight gain in the second and third trimesters recommended in the IOM guidelines is 0.42 kg (0.35-0.50). It would be good to provide a reference for the value of 0,45 kg/week or revise it to be consistent with the IOM guidelines.

3) I find the statistical methods of the study poorly described in general. There is virtually no explanation on how the pseudo-cohort design was reconstructed using the cross-sectional data in the DHS Program, how the total GWG and trimester-specific GWG were computed, or how the pre-pregnancy weight was estimated using the weights of women at risk of conception. I suggest that the statistical analysis and technical details of the methods be much more adequately described.

4) It is not straightforward to combine the survey data across 30 countries into one unified dataset while still preserving the sampling designs (weight, stratification, clustering) appropriate for each dataset. Therefore, I suggest clarifying how the datasets were combined and how stratification and clustering of the original surveys were accounted for. The explanation on the derivation of the new weights using the 2020 population size in the analysis also appears brief to me and will benefit from more details.

5) In Table 2, when classifying women into underweight, normal weight, and overweight or obese, were the WHO cutoffs (i.e., < 18.5, 18.5-<25, >=25) used? For adolescents aged 15 to 19 years old, were the WHO growth references (based on BMI-for-age Z-scores) for children/adolescents used instead of the cutoffs for adults?

6) Page 13: “The overlap in the confidence intervals suggested absence of statistically significant difference between the two groups.” This statement is incorrect in that two CIs may well overlap while still having a statistically significant difference. I suggest the statistical significance in the subgroup analyses be rigorously and quantitatively determined. Otherwise, a qualitative description of the difference may be sufficient without resorting to a discourse on statistical significance.

7) It appears that 11 countries were included in the analysis for 1997-2003, while 30 countries were included in the analysis for 2015-2020. As a result, quite different numbers of countries were used to compare the two time periods (1997-2003 and post-2015). This discrepancy greatly complicates the comparison, and the interpretation of the comparison becomes tricky and unclear.

Reviewer #2: This paper uses DHS data to estimate gestation weight gain in SSA as a whole. It also presents estimates for regions within SSA. I think it is an important project and is well-written and well-presented.

I'd suggest the following major revisions:

1. When you compute the average (body) weight of women who are at risk of conception, I suggest creating and using (statistical) weights that make the non-pregnant group match the age profile of currently pregnant women. The reason to do this is that older women in their 40s with low fecundity may report menstruating and not using contraception, but are unlikely to become pregnant. Yet, (I believe) the current computation considers them equally “at risk of conception” as women in their 20s who are not using contraception. Therefore, their body weights count equally in the average weight of potentially pregnant women. If age is correlated with weight (which it is in many societies), you would be overestimating the average weight of women at risk for pregnancy, which would lead you to underestimate GWG. Both the tables and Figure 1 suggest this is happening because the (body) weights of pre-pregnant women are higher, on average, than the body weights of women in their first trimester. (Although a very small number of women do lose weight in the first trimester, it is not the norm.)

If adjusting for the age profile of potentially pregnant women doesn't bring the pre-pregnancy weight down below first trimester weights, you could try creating statistical weights based on other factors as well (such as number of children or education) but I think age will cover it.

2. I would suggest not including results for women in the first and second months of pregnancy. The women who report very early pregnancy are likely systematically different from the pregnant population, and from those who report 2nd and 3rd trimester pregnancies. I think they will be more educated and perhaps come from less gender-conservative backgrounds. Indeed the fact that such a small fraction of pregnant women report being 1 or 2 months pregnant reinforces the idea that there is under-reporting of pregnancy in these months.

I'd suggest the following minor revisions:

1. The second paragraph of the introduction reads: “In 2019 the Institute of Medicine (IOM) of the National Academies put forth a new guideline on rate of GWG.” It would be good to mention that this guideline is for the US population. If the authors know of any guidelines specific to any SSA country/countries it would be helpful to mention them.

2. Please mention how you code gestational age in the main text. The DHS asks both month pregnant and last period; in some cases the information is missing for one variable and not for the other. Which do you preference? Also, how often is gestational month 10 recorded? You might consider clubbing it with month 9 since it basically descries a full-term pregnancy, but the way that different societies talk about full term pregnancies differs slightly.

3. Please include a more in-depth discussion of the statistical weights in the text.

Reviewer #3: Thank you for the opportunity to review this interesting manuscript. It is an important and interesting topic that can have the potential to improve outcomes for women in sub-Saharan Africa. The paper is well written using clear language. There are, however, some comments I make for your consideration before I believe the paper is ready for publication.

Abstract

1. The authors can add comma after the word ‘On average’ in the result section of the abstract.

2. Please be consistent while reporting mean gestational weight gain with its confidence interval. Sometime you put the mean outside of the bracket and confidence interval inside the bracket while you put all in the bracket in the other time, for example, 6.6 kg (95% confidence interval, 6.0-7.2) vs (10.5, 8.2-12.9 kg)

3. Please be consistent while using ‘kg or kgs’

Background

4. The authors defined Gestational weight gain (GWG) as “the weight increase between conception and just before the birth of the infant.” Could the term ‘baby’ is more appropriate than ‘infant’ here?

5. At the beginning of the second paragraph, the authors stated “In 2019 the Institute of Medicine (IOM) of the National Academies put forth a new guideline on rate of GWG”. I am not sure if the National Academies released a GWG guideline in 2019. Please correct me if I am wrong.

Methods

6. Not clears why and how non-pregnant women were included in the analysis of GWG.

7. I am a bit confused about data inclusion criteria. In the abstract section, the authors stated “Trend in GWG between 2000 and 2015 was determined using the data of 11 SSA countries”. In the method section, the authors stated contradictory statement “In order to estimate mean GWG, we analysed the data … enrolled in DHS implemented in 30 SSA countries since 2010.” At the same time, the authors included data collected since 1997. Although the authors reported that they used DHS data since 2010 to estimate mean GWG, the finding on the figure two showed that the authors estimated mean GWG since 1997. The authors may need to clarify more about this.

Result and discussion

8. Can the authors estimate GWG for underweight, normal weight, over weight and obese women separately? For example, if we see their GWG estimate for Southern African sub-region, 10.5 kgs (95% CI: 8.2-12.9), this value is excessive for obese women; adequate for overweight women; and inadequate for normal weight and underweight women. Therefore, the GWG estimates in this paper are difficult to interpret unless stratified based on pre-pregnancy body mass index of the women. These findings tell us nothing about the adequacy of GWG in the way that the authors reported. In the discussion section, the authors interpreted GWG as “we estimated (6.6 kg) is only about half of the minimum recommended gain for women with normal baseline BMI”. However, authors didn’t tell us whether 6.6 kg was estimated for normal weight women or obese women. The meanings and implications of this finding, GWG of 6.6kg, is totally different for normal weight or obese women.

Conclusion

The authors concluded that “The mean GWG in SSA is very low”. However, the nature of their result do not allow them to conclude so.

6. PLOS authors have the option to publish the peer review history of their article (what does this mean?). If published, this will include your full peer review and any attached files.

Reviewer #1: No

Reviewer #2: No

Reviewer #3: No

---

## [Author Response · Author response to Decision Letter 0]

20 Mar 2021

Comments from the Academic Editor 

Comment 1: With respect to the introduction, the paper should provide background that puts the manuscript into context and allows readers outside the field to understand the purpose and significance of the study and include a brief review of the key literature. In the present case, as reviewer 1 points out, the article is missing a key reference that looks at the same topic with the same data and similar methods but for more countries. It is basic in this case to note the differences in approach or in scope. Note that otherwise PLOS ONE policy states that if a submitted study replicates or is very similar to previous work, authors must provide a sound scientific rationale for the submitted work and clearly reference and discuss the existing literature. Submissions that replicate or are derivative of existing work will likely be rejected if authors do not provide adequate justification.

Response: The concern is right. The key literature (Wang et al., 2020) identified by the first reviewer was not included because it was published after the submission of this manuscript. Now we have cited this article and described its findings both in the introduction and discussion sections. Further we have presented how our study is different from this article, including our peculiar contribution to the existing body of literature on the topic. It is important to note that the study by Wang et al., has not provided information the following information that we did: (1) trends in GWG in SSA, (2) sub-national estimates with SSA, and (3) differences in GWG trajectories across basis maternal socio-demographic characteristics. The same is now stated in the fourth paragraph of the Introduction section (Page 5). Furthermore, to allow readers outside the field to understand the purpose and significance of the study, we have now strengthened the last paragraph of the introduction section.

Comment 2: Regarding methods, there are some limitations that should be addressed. A prior consideration, is that the methods should be adequately described to allow for replication. As noted by reviewer 1, this is not the case, and the missing information might have confused, for instance, reviewer 3 in understanding the method.

Response: Additional methodological descriptions are now provided at different parts of the section. 

Comment 3: First, note the concerns regarding the use of relevant guidelines as suggested by reviewers 1, 2 and 3.

Response: To the best of our knowledge, there is no GWG guideline specific to the SSA region or a country in this region. It is important to note that, even though the IOM guideline was developed for US population, its applicability to other settings had been somehow validated (Goldstein et al. 2018; Nomura et al. 2019; and Wie et al. 2017). The same is now stated in the second and third paragraphs of the introduction section. The absence of SSA-specific guideline is now discussed in the discussion section. 

Comment 4: The problem of the relevant comparison group raised by reviewers 2 and 3. Reviewer 2 suggestion of reweighting non-pregnant women to match pregnant women according to weight is interesting and doable. If you carry this out show, at least for some instances, both sets of estimates to see if the strange patterns for first trimester disappear. Alternatively, a more sophisticated approach would be the use of propensity score matching to ensure that women in the synthetic control group (nonpregnant) are comparable to those pregnant.

Response: Reviewer 2 recommended for adjusting pregnant and non-pregnant women by age. As we tried to respond to the reviewer below, theoretically non-pregnant women are likely to be older and tend to have higher weight due to postpartum weight retention. Yet, in our dataset there was no meaningful difference in the age profile of non-pregnant women (26.9 ± 10.1 years) and women in the first (26.6 ± 6.9 years), second (26.8 ± 6.7 years), and third (26.9 ± 6.5 years) trimesters. Accordingly, we felt that additional age adjustment does not add value to the validity of the estimation. 

Comment 5: Comment by reviewer 3 regarding grouping by weight I don’t believe it is feasible since you do not observe prepregnancy weight. However, you should bear it in mind when analyzing the patterns of estimated gestational weight gain. Since you have computed the proportions underweight and overweight you could see if the pattern of estimated weight gain at the survey level correlate with the proportions underweight and overweight. That will be very helpful for the discussion and also to tease out possible factors behind the much smaller GWG in Africa compared to other developing regions according to the missing source noted by reviewer 1.

Response: Inability to stratify GWG based on prepregnancy BMI or weight is now discussed as a limitation of the study and the possibility of over or underestimation of GWG is also stated (Discussion section, sixth paragraph, page 19). It is important to note that most of the limitations of this study can lead to both under or overestimation of GWG. 

Comment 6: Third, as noted by reviewer 2, not using data from women in the first two months of pregnancy is probably a good idea. For instance, it is known that DHS patterns of reported pregnancy termination are extremely low in most sub-Saharan African countries (https://doi.org/10.1371/journal.pone.0221178) suggesting that many early terminations are missing because the woman was unaware of the pregnancy. Since this proportion would vary from survey to survey and according to socioeconomic characteristics, it would induce a bias in estimate weight gain.

Response: Please see the response we have provided to the reviewer below. Thank you. 

Comment 7: In your revision note that PLOS ONE endorses the use of the STROBE checklist (http://www.strobe-statement.org) for observational studies such as this one. Make sure that all the questions in the checklist are addressed.

Response: We have completed the STROBE checklist and provided as a supporting file. 

 

Reviewer #I

Comment 1: This work is similar to a previous study in terms of the research question. In Wang et al. (Gestational weight gain in low-income and middle-income countries: a modelling analysis using nationally representative data. BMJ global health. 2020), a multilevel modelling framework was applied to the DHS data to estimate the mean GWG in the year 2015 in all low- and middle-income countries and regions of the world, including in sub-Saharan Africa. Surprisingly, the Wang et al. paper was not referenced or mentioned in this work. Discussing the similarities and differences in the results and methodology between the two works will be necessary.

Response: Thank you for proposing this important paper. Earlier we did not cite this paper because it was not yet published at the time of this manuscript was submitted to this journal. Now the findings of the study are discussed and compared (Discussion Section, Paragraph 3, page 17) and what the current study would add in light with the new literature are described (Introduction Section, Paragraph 4). 

Comment 2: On page 4: “assuming 0.5-2 kg gain in the first and 0.45 kg/week in the last two trimesters." However, the weekly weight gain in the second and third trimesters recommended in the IOM guidelines is 0.42 kg (0.35-0.50). It would be good to provide a reference for the value of 0,45 kg/week or revise it to be consistent with the IOM guidelines.

Response: Thank you. We have now corrected it as 0.42 kg/week.

Comment 3: I find the statistical methods of the study poorly described in general. There is virtually no explanation on how the pseudo-cohort design was reconstructed using the cross-sectional data in the DHS Program, how the total GWG and trimester-specific GWG were computed, or how the pre-pregnancy weight was estimated using the weights of women at risk of conception. I suggest that the statistical analysis and technical details of the methods be much more adequately described.

Response: In the data management and analysis section (Page 10), we have now described how GWG and trimester specific GWG were computed. In page 9, we have now stated how GWG trajectories were reconstructed based on cross-sectional data. 

Comment 4: It is not straightforward to combine the survey data across 30 countries into one unified dataset while still preserving the sampling designs (weight, stratification, clustering) appropriate for each dataset. Therefore, I suggest clarifying how the datasets were combined and how stratification and clustering of the original surveys were accounted for. The explanation on the derivation of the new weights using the 2020 population size in the analysis also appears brief to me and will benefit from more details.

Response: Merging the datasets was not a major problem in this study because all the DHS surveys used similar sampling scheme, sampling weight calculation approach and variable names. What we did was adjusting the existing sample weights for the population size of the countries (i.e. poststratification weighting). Regarding poststratification weighting, we have now provided an additional explanation on how we calculated it based on the population size of the countries in the year 2020. 

Comment 5: In Table 2, when classifying women into underweight, normal weight, and overweight or obese, were the WHO cutoffs (i.e., < 18.5, 18.5-<25, >=25) used? For adolescents aged 15 to 19 years old, were the WHO growth references (based on BMI-for-age Z-scores) for children/adolescents used instead of the cutoffs for adults?

Response: Thank you for this interesting comment. We have now provided BMI values only for adults 19 years or above based on the usual BMI cut values. 

Comment 6: Page 13: “The overlap in the confidence intervals suggested absence of statistically significant difference between the two groups.” This statement is incorrect in that two CIs may well overlap while still having a statistically significant difference. I suggest the statistical significance in the subgroup analyses be rigorously and quantitatively determined. Otherwise, a qualitative description of the difference may be sufficient without resorting to a discourse on statistical significance.

Response: We disagree with comments of the reviewers. In general, examining statistical difference based on overlap of confidence intervals is a simple and convenient approach specially when graphs of confidence intervals have been presented (Schenker & Gentleman, 2001: On judging the significance of differences by examining the overlap between confidence intervals). Specially in this manuscript, relying on overlap of confidence intervals is the only option because we did not have individual level data to estimate p-values for comparing the distribution of two or more means. However, it is important to note that the method of examining overlap can lead to more type II errors than the standard p-values and it mistakenly accepts the null hypothesis when p-values show marginally significant differences (Schenker & Gentleman, 2001; Austin & Hux , 2002). This limitation is now discussed in the ninth paragraph of the discussion section (Page 20). 

Comment 7: It appears that 11 countries were included in the analysis for 1997-2003, while 30 countries were included in the analysis for 2015-2020. As a result, quite different numbers of countries were used to compare the two time periods (1997-2003 and post-2015). This discrepancy greatly complicates the comparison, and the interpretation of the comparison becomes tricky and unclear.

Response: Seems there is misunderstanding here. As we tried to describe at the end of the introduction section (Page 5), this study had two objectives (i) estimate mean GWG in SSA based on aggregated data from multiple nationally representative cross-sectional surveys and; (ii) to compare changes in mean GWG between 2000 and 2015 in the region. For the first objective, the data of 30 countries that implemented DHS between 2015-2020 were included. For the second objective the data of 11 countries that conducted DHS surveys between 1997-2003 and 2015-2020 analysed. So, the trend analysis was conducted based on the data of 11similar set of countries. This is stated in the second and third paragraphs of the “Data source and inclusion criteria section” (Page 6)

 

Reviewer #2

Comment 1: When you compute the average (body) weight of women who are at risk of conception, I suggest creating and using (statistical) weights that make the non-pregnant group match the age profile of currently pregnant women. The reason to do this is that older women in their 40s with low fecundity may report menstruating and not using contraception, but are unlikely to become pregnant. Yet, (I believe) the current computation considers them equally “at risk of conception” as women in their 20s who are not using contraception. Therefore, their body weights count equally in the average weight of potentially pregnant women. If age is correlated with weight (which it is in many societies), you would be overestimating the average weight of women at risk for pregnancy, which would lead you to underestimate GWG. Both the tables and Figure 1 suggest this is happening because the (body) weights of pre-pregnant women are higher, on average, than the body weights of women in their first trimester. (Although a very small number of women do lose weight in the first trimester, it is not the norm.) If adjusting for the age profile of potentially pregnant women doesn't bring the pre-pregnancy weight down below first trimester weights, you could try creating statistical weights based on other factors as well (such as number of children or education) but I think age will cover it.

Response: Thank you for this interesting comment. Theoretically the suggestion of the reviewer is right. yet, in our data there was no meaningful difference in the age profile of non-pregnant women (26.9 ± 10.1 years) and women in the first (26.6 ± 6.9 years), second (26.8 ± 6.7 years), and third (26.9 ± 6.5 years) trimesters. Accordingly, we felt that additional age adjustment does not add value to improve the comparability of non-pregnant and pregnant women. 

Comment 2: I would suggest not including results for women in the first and second months of pregnancy. The women who report very early pregnancy are likely systematically different from the pregnant population, and from those who report 2nd and 3rd trimester pregnancies. I think they will be more educated and perhaps come from less gender-conservative backgrounds. Indeed, the fact that such a small fraction of pregnant women report being 1 or 2 months pregnant reinforces the idea that there is under-reporting of pregnancy in these months.

Response: It is true that the small number of women in the first and second months indicates the presence of under reporting of pregnancy in early pregnancy. It is also theoretically true that women who are aware of pregnancy as early as first or second gestational month are likely to be more educated or come from better off households. However, we don’t think this has affected our primary objectives (estimating average and trimester-specific GWG and determining recent trends in GWG) for the following reasons. (1) GWG is determined as a difference between pre-pregnancy weight and weight at the time of delivery. As a result, imprecision or errors in weight in the first one or two months cannot affect this estimate. (2) Trimester-specific GWG estimates are made based on the maternal weight at the third, sixth and end of pregnancy weights and it is not dependent on the weight on the first one or two months. (3) Practically we did not observe meaningful difference between women in the first two gestational months and women in their third or higher gestational duration in terms of multiple socio-demographic factors including being in lowest wealth quantile (22.0% vs 20.6%), having only primary level of education (34.8% vs 32.7%), parity of 2 or above (21.5% vs 24.9%), age less than 20 years (14.3% vs 13.9%); and, (3) while presenting weight gain trajectories graphically, we did not observe a different or unexpected pattern in those two months. 

Comment 3: The second paragraph of the introduction reads: “In 2019 the Institute of Medicine (IOM) of the National Academies put forth a new guideline on rate of GWG.” It would be good to mention that this guideline is for the US population. If the authors know of any guidelines specific to any SSA country/countries it would be helpful to mention them.

Response: The concern raised is right. We have now stated in the second paragraph of the Introduction section that the IOM is US-specific guideline however studies reported that it is applicable in other settings (especially in the Asian continent). The applicability of the IOM guideline had not been validated in Africa setting and to the best of our knowledge there is no SSA-specific gestational weight gain guideline. This is now discussed as a limitation of the study in the seventh paragraph of the discussion section (Page 18). 

Comment 4: Please mention how you code gestational age in the main text. The DHS asks both month pregnant and last period; in some cases, the information is missing for one variable and not for the other. Which do you preference? Also, how often is gestational month 10 recorded? You might consider clubbing it with month 9 since it basically descries a full-term pregnancy, but the way that different societies talk about full term pregnancies differs slightly.

Response: The following information is now given under the “Data management and analysis” sub-section: “When available, gestational age was determined based on LNMP otherwise it was estimated based on self-report of the women. Very small proportion (about 0.3%) of pregnant women reported gestational age of 10 months and during analysis it was recoded to 9 months.”

Comment 5: Please include a more in-depth discussion of the statistical weights in the text.

Response: Additional information is now provided in the second paragraph of the data analysis section (page 9-10) on how the sample weighting was made.

 

Reviewer #3

Comment 1: Abstract. The authors can add comma after the word ‘On average’ in the result section of the abstract. Please also be consistent while reporting mean gestational weight gain with its confidence interval. Sometimes you put the mean outside of the bracket and confidence interval inside the bracket while you put all in the bracket in the other time, for example, 6.6 kg (95% confidence interval, 6.0-7.2) vs (10.5, 8.2-12.9 kg). Please be consistent while using ‘kg or kgs’

Response: corrected.

Comment 2: Background: The authors defined Gestational weight gain (GWG) as “the weight increase between conception and just before the birth of the infant.” Could the term ‘baby’ is more appropriate than ‘infant’ here.

Response: corrected.

Comment 3: At the beginning of the second paragraph, the authors stated “In 2019 the Institute of Medicine (IOM) of the National Academies put forth a new guideline on rate of GWG”. I am not sure if the National Academies released a GWG guideline in 2019. Please correct me if I am wrong.

Response: Apologies for this silly error. We have now changed 2019 to 2009. 

Comment 4: Not clears why and how non-pregnant women were included in the analysis of GWG.

Response: As we tried to describe in the study design of the methods section (Page 6, first paragraph) pre-pregnancy weight was estimated using the data of non-pregnant women at risk of conception. Pre-pregnancy weight can not be determined only based on the data of pregnant women, because this may overestimate the baseline weight and underestimate the GWG.

Comment 5: I am a bit confused about data inclusion criteria. In the abstract section, the authors stated “Trend in GWG between 2000 and 2015 was determined using the data of 11 SSA countries”. In the method section, the authors stated contradictory statement “In order to estimate mean GWG, we analysed the data … enrolled in DHS implemented in 30 SSA countries since 2010.” At the same time, the authors included data collected since 1997. Although the authors reported that they used DHS data since 2010 to estimate mean GWG, the finding on the figure two showed that the authors estimated mean GWG since 1997. The authors may need to clarify more about this.

Response: In general, our intention was to assess the trends in GWG between 2000 and 2015. However, nearly all countries do not have data in these specific years. However, they have conducted surveys around 2000 and 2015. Accordingly, we have included surveys conducted 

around the beginning of the new millennium (1997-2003) and on or after 2015 (2015-2018). This may marginally over or underestimate the rate of GWG. We have now stated this as a limitation of the study in the eighth paragraph of the discussion section (Page 19).

Comment 6: Result and discussion: Can the authors estimate GWG for underweight, normal weight, overweight and obese women separately? For example, if we see their GWG estimate for Southern African sub-region, 10.5 kgs (95% CI: 8.2-12.9), this value is excessive for obese women; adequate for overweight women; and inadequate for normal weight and underweight women. Therefore, the GWG estimates in this paper are difficult to interpret unless stratified based on pre-pregnancy body mass index of the women. These findings tell us nothing about the adequacy of GWG in the way that the authors reported. In the discussion section, the authors interpreted GWG as “we estimated (6.6 kg) is only about half of the minimum recommended gain for women with normal baseline BMI”. However, authors didn’t tell us whether 6.6 kg was estimated for normal weight women or obese women. The meanings and implications of this finding, GWG of 6.6kg, is totally different for normal weight or obese women.

Response: The concern of the reviewer is right however it was not possible to do stratification by BMI because BMI was only available for pregnant women. BMI was only available for non-pregnant women and hence reconstructing GWG for different BMI categories was not possible. This is discussed as a limitation in the seventh paragraph of the discussion section (Page 19).

Comment 7: Conclusion: The authors concluded that “The mean GWG in SSA is very low”. However, the nature of their result do not allow them to conclude so.

Response: We disagree with this comment of the reviewer. In the results section we have demonstrated the mean GWG in SSA is only 6.6kg and this is largely sub-optimal as compared to the IOM guideline. According to IOM, pregnant women with normal pre-pregnancy BMI should gain 11.5-16.0 kg of weight and women with low pre-pregnancy BMI are expected to gain as high as 18 kg of weight. So, we think that it is reasonable to conclude that the mean GWG in SSA is very low.

---

## [Decision Letter · Decision Letter 1]

12 May 2021

Gestational weight gain in sub-Saharan Africa: Estimation based on pseudo-cohort design

PONE-D-21-02601R1

Dear Dr. Gebremedhin,

We’re pleased to inform you that your manuscript has been judged scientifically suitable for publication and will be formally accepted for publication once it meets all outstanding technical requirements.

Kind regards,

José Antonio Ortega, Ph.D.

Academic Editor

PLOS ONE

Additional Editor Comments (optional):

[staff editor's edits]

The suggestion [from reviewer 2] to reweight the observations is possible to carry out. In statistical terms doing so would reduce bias at the cost of increasing variance. In the first revision I suggested the authors to consider the idea, which I believe is good at least to try. They preferred not do so and argued regarding balance on age. The reviewer argues that there could be second-order effects connected to variance and there could be lack of balance on other dimensions. There are others instances of published research where this has not been done and I believe the authors should have some leeway, particularly when considering that it cannot be said that doing so will always improve the estimation. If we take a mean squared error criterion it would depend on how large is the bias to see if it compensates for the increased variance.

Based on these considerations, I feel that the article fulfills PLOS ONE publication criteria and that this particular choice is optional for the authors who ultimately are taking responsibility for their published work.

Reviewers' comments:

Reviewer's Responses to Questions

**Comments to the Author**

1. If the authors have adequately addressed your comments raised in a previous round of review and you feel that this manuscript is now acceptable for publication, you may indicate that here to bypass the “Comments to the Author” section, enter your conflict of interest statement in the “Confidential to Editor” section, and submit your "Accept" recommendation.

Reviewer #1: All comments have been addressed

Reviewer #2: (No Response)

Reviewer #3: All comments have been addressed

2. Is the manuscript technically sound, and do the data support the conclusions?

Reviewer #1: Yes

Reviewer #2: Partly

Reviewer #3: Partly

3. Has the statistical analysis been performed appropriately and rigorously? 

Reviewer #1: Yes

Reviewer #2: No

Reviewer #3: I Don't Know

4. Have the authors made all data underlying the findings in their manuscript fully available?

Reviewer #1: Yes

Reviewer #2: Yes

Reviewer #3: Yes

5. Is the manuscript presented in an intelligible fashion and written in standard English?

Reviewer #1: Yes

Reviewer #2: Yes

Reviewer #3: Yes

6. Review Comments to the Author

Reviewer #1: (No Response)

Reviewer #2: I suggested that the authors make adjustments to the weights of non-pregnant women to obtain a better estimate of pre-pregnancy weight. The fact that non-pregnant women weight more than first trimester pregnant women is a sign that there is selection into pregnancy that is correlated with weight. I suggested age would be a good place to start looking for factors on which to adjust. The authors did not show additional analyses to respond to this comment. The fact the average age is similar between pregnant and non-pregnant women is similar does not tell us that no adjustment is needed. The distribution of age may be different. There may also be other characteristics (such as number of children, education, asset wealth) that could be used to provide a better estimate of pre-pregnancy weight, a quantity which is at the heart of the authors' analysis.

Further, the authors say that they don't include results for women who are 1 and 2 months pregnant, but Fig 1 does include these women.

I would continue to suggest that the authors give more thought into how selection into pregnancy and pregnancy reporting affects their analysis, and write up their results in such a way as to reflect this. As is, the paper continues to assume that non-pregnant and pregnant women of each gestational age are similar, without providing evidence for these assumptions in the paper.

Reviewer #3: I would like to thank the authors for addressing almost all of my comments expect the one they disagree with. Although the authors disagreed with my last comment, their argument is still not convincing. As the authors rightly stated, the mean gestational weight gain, 6.6kg, was low for normal weight women according to IOM recommendations. However, the mean weight gain of 6.6kg is adequate for obese women given they are recommended to gain 5 to 9kg. Moreover, the authors clearly stated that they do not have data regarding BMI for pregnant women. This means that they are unable to tell whether the mean weight gain of 6.6kg is either for normal weight women or obese women. The mean weight gain of 6.6kg is low for normal weight women but it is adequate for obese women. Anyway, it is up to you and the editor to decide on if convinced on your argument.

Wish you all the best

7. PLOS authors have the option to publish the peer review history of their article (what does this mean?). If published, this will include your full peer review and any attached files.

Reviewer #1: No

Reviewer #2: No

Reviewer #3: No

---

## [Editor Report · Acceptance letter]

17 May 2021

PONE-D-21-02601R1 

Gestational weight gain in sub-Saharan Africa: Estimation based on pseudo-cohort design 

Dear Dr. Gebremedhin:

I'm pleased to inform you that your manuscript has been deemed suitable for publication in PLOS ONE. Congratulations! Your manuscript is now with our production department. 

Kind regards, 

on behalf of

Dr. José Antonio Ortega 

Academic Editor

PLOS ONE